# Effectiveness of Using the FreeStyle Libre^®^ System for Monitoring Blood Glucose during the COVID-19 Pandemic in Diabetic Individuals: Systematic Review

**DOI:** 10.3390/diagnostics13081499

**Published:** 2023-04-21

**Authors:** Luelia Teles Jaques-Albuquerque, Elzi dos Anjos-Martins, Luiza Torres-Nunes, Ana Gabriellie Valério-Penha, Ana Carolina Coelho-Oliveira, Viviani Lopes da Silva Sarandy, Aline Reis-Silva, Adérito Seixas, Mario Bernardo-Filho, Redha Taiar, Danúbia Cunha de Sá-Caputo

**Affiliations:** 1Laboratório de Vibrações Mecânicas e Práticas Integrativas, Departamento de Biofísica e Biometria, Instituto de Biologia Roberto Alcântara Gomes, Policlínica Universitária Piquet Carneiro, Universidade do Estado do Rio de Janeiro, Rio de Janeiro 20950-003, Brazil; lueliaa19@gmail.com (L.T.J.-A.); elzireis@gmail.com (E.d.A.-M.); ltnmamae@gmail.com (L.T.-N.); anagabriellie.vpenha@gmail.com (A.G.V.-P.); anacarol_coelho@hotmail.com (A.C.C.-O.); vivisarandy@gmail.com (V.L.d.S.S.); fisio.alinereis@hotmail.com (A.R.-S.); bernardofilhom@gmail.com (M.B.-F.);; 2Mestrado Profissional em Saúde, Medicina Laboratorial e Tecnologia Forense, Universidade do Estado do Rio de Janeiro, Rio de Janeiro 20950-003, Brazil; 3Programa de Pós-Graduação em Fisiopatologia Clínica e Experimental, Universidade do Estado do Rio de Janeiro, Rio de Janeiro 20950-003, Brazil; 4Programa de Pós-Graduação em Ciências Médicas, Universidade do Estado do Rio de Janeiro, Rio de Janeiro 20950-003, Brazil; 5Escola Superior de Saúde Fernando Pessoa, 4200-256 Porto, Portugal; aderito@ufp.edu.pt; 6MATériaux et Ingénierie Mécanique (MATIM), Université de Reims Champagne Ardenne, 51100 Reims, France

**Keywords:** COVID-19, artificial intelligence, pandemic, type 2 diabetes mellitus, social isolation

## Abstract

Background: Artificial Intelligence (AI) is an area of computer science/engineering that is aiming to spread technological systems. The COVID-19 pandemic caused economic and public health turbulence around the world. Among the many possibilities for using AI in the medical field is FreeStyle Libre^®^ (FSL), which uses a disposable sensor inserted into the user’s arm, and a touchscreen device/reader is used to scan and retrieve other continuous monitoring of glucose (CMG) readings. The aim of this systematic review is to summarize the effectiveness of FSL blood glucose monitoring during the COVID-19 pandemic. Methods: This systematic review was carried out in accordance with the Preferred Reporting Items for Systematic Review and Meta-Analyses (PRISMA) and was registered in the international prospective register of systematic reviews (PROSPERO: CRD42022340562). The inclusion criteria considered studies involving the use of the FSL device during the COVID-19 pandemic and published in English. No publication date restrictions were set. The exclusion criteria were abstracts, systematic reviews, studies with patients with other diseases, monitoring with other equipment, patients with COVID-19, and bariatrics patients. Seven databases were searched (PubMed, Scopus, Embase, Web of Science, Scielo, PEDro and Cochrane Library). The ACROBAT-NRSI tool (A Cochrane Risk of Bias Assessment Tool for Non-Randomized Studies) was used to evaluate the risk of bias in the selected articles. Results: A total of 113 articles were found. Sixty-four were excluded because they were duplicates, 39 were excluded after reading the titles and abstracts, and twenty articles were considered for full reading. Of the 10 articles analyzed, four articles were excluded because they did not meet the inclusion criteria. Thus, six articles were included in the current systematic review. It was observed that among the selected articles, only two were classified as having serious risk of bias. It was shown that FSL had a positive impact on glycemic control and on reducing the number of individuals with hypoglycemia. Conclusion: The findings suggest that the implementation of FSL during COVID-19 confinement in this population can be confidently stated to have been effective in diabetes mellitus patients.

## 1. Introduction

Health technologies have advanced over the years [1]. Artificial Intelligence is a branch of computer science/engineering/technology that aims to develop computer systems to perform tasks that can be performed better by humans than by machines, or which do not have a viable algorithmic solution in conventional computing [2]. AI is used in medicine using computers that, by analyzing a large volume of information and following algorithms defined by experts, can propose solutions to medical problems [3,4].

In a situation of an epidemiological outbreak, in which a disease spreads easily and quickly, such as the 2019 coronavirus outbreak, the speed and efficiency of information is of paramount importance. As a consequence, AI is fundamental, because humans are not able to analyze large amounts of data and information with the necessary speed and efficiency [4].

The COVID-19 pandemic generated serious global economic and public health problems [5,6,7]. Severe Acute Respiratory Syndrome Coronavirus 2 (SARS-CoV-2) was first recorded in the city of Wuhan in China, and spread rapidly around the world. It is highly contagious [8,9]. The World Health Organization (WHO) declared a Public Health Emergency of International Concern (ESPII) on 30 January 2020, and designated the COVID-19 outbreak as a pandemic on 11 March 2020, because of the increased rates of COVID-19 in various countries and regions of the world [10,11,12]. In Brazil, SARS-CoV-2 impaired health (in the most severe cases), resulting in worse prognoses, especially in groups considered to be at high risk, e.g., elderly people over 60 years of age, the immunocompromised, cardiopaths, people with lung disease, diabetics, and patients with chronic comorbidities [13,14].

The use of AI during the COVID-19 pandemic was effective, and this has been highlighted [15,16,17,18]. Among the various possibilities of using AI in the medical field is FreeStyle Libre^®^ (FSL) [19]. FSL is related to AI systems, and as the advantage of being able to help individuals suffering from Type 1 (T1DM) or Type 2 (T2DM) diabetes mellitus to achieve better glucose control, because it does not require finger-prick calibration. Strict blood glucose control is effective at reducing the risk of long-term complications from diabetes mellitus (DM), such as retinopathy, nephropathy, neuropathy, coronary heart disease, ischemic stroke, and peripheral vascular disease [20]. It is important to consider that AI is not a solution, but a relevant and alternative tool to assist in the solution [21]. The purpose of AI is to guide a response, improve care, and save lives [22]. AI was seen in the context of the COVID19 pandemic as a “force multiplier”, since the world was facing a great challenge, in which it was necessary to carry out large-scale and short-term activities [4]. 

FSL is a simple device, and consists of two components: a disposable sensor, which is inserted into the user’s upper arm, and a touchscreen device/reader, which is used to scan and retrieve other continuous glucose monitoring (CGM) readings [23]. FSL has been an important tool for patients with T2DM [24], with additional advantages for this population, which is considered a high-risk group for infectious diseases, whereby individuals with insulin resistance have a 50–60% higher risk of pulmonary infection [25].

To achieve adequate quality of life and reduce long-term problems, patients are increasingly being encouraged to take an active role in managing their condition. Therefore, proper treatment management, aimed at tight blood glucose control, reduces the risk of long-term complications of DM [26]. In September 2016, the Food and Drug Administration (FDA) approved this equipment for glucose monitoring, CGM, for professional use in clinics, and in September 2017, the FDA approved the equipment for personal use [27].

The system, unlike others, does not require finger-prick calibration, as this functionality is built into the core technology [28]. However, unlike other systems, it is necessary for the individual to take active steps to gain access to real-time glucose values by moving the receiver over the sensor. It is also available to patients from pharmacies, and unlike other CGM technologies on the market, where the time between the patient’s decision and delivery can be months, FSL does not require the involvement of specialized distributors, speeding up the process [29]. The FSL is also the only CGM on the market to date that does not interfere with acetaminophen, unlike other CGM technologies, which provide artificially high glucose readings in patients taking acetaminophen [30]. A remarkable improvement in DM-related distress and clinical parameters has been reported in patients switching from finger-prick monitoring to flash glucose monitoring systems [31]. FSL improved the patients’ DM levels, as well as increasing metabolic glycemic control [32]. In general, the use of the AI has several advantages; however, the limitations of this technology must be considered. 

The present systematic review aims to summarize the effectiveness of using FSL blood glucose monitoring during the COVID-19 pandemic.

## 2. Materials and Methods 

This systematic review was carried out in accordance with the recommendations of the Preferred Reporting Items for Systematic Reviews and Meta-Analyses (PRISMA) [33] and was registered on the International Prospective Register of Systematic Reviews (PROSPERO: CRD42022340562).

### 2.1. Eligibility Criteria

Studies were included if they (i) involved the use of the FSL device during the COVID-19 pandemic, (ii) included people with DM during the COVID-19 pandemic, and (iii) were published in the English language. No publication date restrictions were set. Exclusion criteria were congress abstracts, systematic reviews, patients with other diseases associated with T2DM or T1DM, monitoring with other equipment (i.e., not an FSL device), patients with COVID-19, and bariatrics patients. 

### 2.2. Operational Settings

FSL is a sensor that was developed to allow frequent measurements of an individual’s glucose level. These measurements are minimally invasive and instantaneous. This sensor is well tolerated and adopted by patients due to its ease of use, small size and lower cost compared to its competitors. The FSL sensor is factory calibrated, reducing the need to determine blood glucose number by fingerstick [24]. 

### 2.3. Search Strategy

The electronic search was performed in the PubMed, Scopus, Embase, Web of Science, Scielo, PEDro and Cochrane Library databases using the following search string “Freestyle libre” and “COVID” on 24 February 2023. Searches in PEDro resulted in zero articles. The keywords used in the search were defined in the PICO strategy, focusing on diabetic patients and patients with complications from COVID-19 (Participants), FSL (Exposure), without comparison (Comparison), glycemic control (Outcome), in order to answer the question: what was the effectiveness of using FSL to monitor blood glucose during the COVID-19 pandemic in diabetic individuals?

### 2.4. Selection of Studies and Data Extraction

All references were exported to software (Microsoft Word) and duplicates were removed. The current review was carried out by following 4 steps. Records were identified through a database search and reference screening (Identification), two reviewers (LN and LTJA) independently examined the titles and abstracts, and irrelevant studies were excluded based on eligibility criteria (Screening). Differences were resolved through discussion with a third reviewer (AGVP). Relevant full articles were analyzed for eligibility (Eligibility), and all studies that met the eligibility criteria were included in the systematic review. There was no disagreement between the authors. The same researchers were responsible for independently extracting data from the included studies. Data related to the study information (author and year), subjects, place of allocation, period, objective, results/conclusion, study design and level of evidence were extracted for presentation in this systematic review.

### 2.5. Level of Evidence

Figure 1 shows the hierarchy scale from the National Health and Medical Research Council that was used to access the level of evidence of each of the selected publications [34]. 

### 2.6. Methodological Quality of the Studies

Three reviewers (LTJA, LTN and AGVP) determined the risk of bias of the included studies using the ACROBAT-NRSI tool (A Cochrane Risk of Bias Assessment Tool for Non-Randomized Studies) [35], comparing the health effects of two or more interventions. This tool covers seven domains, which are divided chronologically into pre-intervention, intervention, and post-intervention. Each item is classified as being at low, moderate, severe, or critical risk of bias. It is necessary to state when no information was present. A general judgment was made of the risk of bias on the basis of an assessment of the individual domains, with the most cited rating prevailing. However, in practice, some ‘severe’ risks of bias (or ‘moderate’ risks of bias) may be considered additive, so that risks of bias in various domains can lead to an overall judgment of a ‘serious’ risk of bias. After analyzing the seven domains, the overall risk of bias (RoB) judgment was placed in the eighth domain. Additionally, the ROBINS-I tool, a development of the ACROBAT-NSRI tool, was used to display the risk of bias, with each being represented by colors [36]. 

## 3. Results

A total of 113 articles were found in the PubMed, Scopus, Embase, Web of Science, Scielo, Cochrane Library and PEDro databases. Of these, 64 were excluded because they were duplicates. After reading titles and abstracts, 39 articles were excluded and 10 articles were considered for full reading. Of the 10 articles analyzed, four articles were excluded because they did not match the inclusion criteria. Thus, six articles were included in this systematic review, as shown in the flowchart in Figure 2.

Table 1 presents the main characteristics of the studies included in the current systematic review, including author/year/country, subjects, place of allocation, period, objective, results/conclusion, study design and level of evidence.

Di Dalmazi et al., 2020 [37] reported that CGM metrics were improved mostly in children and adults, while they remained unchanged in adolescents. Bonora et al., 2020 [38] observed that glycemic control improved in T1DM patients who stopped working during the block, suggesting that slowing down routine daily activities may have beneficial effects on T1DM management, at least in the short term. Navis et al., 2020 [39] reported that sensor-based glycemic outcomes in people with T1DM in the current cohort improved during COVID-19 lockdown. Dexcom G6 (DG6) had a shorter % time (<3.9 mM) compared to FSL. Dover et al., 2020 [40] described that there was a small reduction in the number of individuals with hypoglycemia. Blockade was not associated with a deterioration in glycemic control in people with T1DM using flash glucose monitoring. Cervantes-Torres and Romero-Blanco, 2022 [41] reported that their analyses showed differences related to the use of the sensor. After the study period, patients obtained better levels of basal glucose, glycosylated haemoglobin, creatinine, cholesterol, triglycerides and LDL. Choudhary et al., 2022 [42] reported that in January, prior to the COVID-19 pandemic, the 65 years or older age group had the highest %TIR (57.9%), while the 18–25 years age group had the lowest (51.2%) (*p* < 0.001). 

Figure 3 shows the assessment of the risk of bias of the studies included in this systematic review, determined according to the A Cochrane Risk of Bias Assessment Tool for Non-Randomized Studies (ACROBAT-NRSI). It can be observed that, among the selected articles, one study is classified as having low risk of bias, three are classified as being at moderate risk of bias, and two are classified as being at serious risk of bias.

## 4. Discussion

The main objective of this systematic review was to assess the effectiveness of using FSL blood glucose monitoring during the COVID-19 pandemic. After analyzing the included studies, the results suggest that the use of FSL showed a positive impact on glycemic control and on reducing the number of individuals with hypoglycemia. However, the shortcomings in the methodological quality of the included studies were serious, particularly regarding confounding bias, bias due to selection of participants and bias due to missing data. This shows that the articles included in general contain important errors that could interfere with their methodological validity.

During the lockdown period, many outpatient services were interrupted, including in the case of DM and other specializations, who were moved to provide hospital care in the face of COVID-19 [43]. In addition, due to social isolation, many services have moved to the remote format, and telemedicine had gained space to provide medical care to patients, especially those with comorbidities and risk of deterioration due to COVID-19. In this context, in association with the difficulty of accessing health services and the fear of contamination, there was a significant drop in the monitoring of disease, including DM [44].

Another change in the lockdown period was related to routine, where a suspension of face-to-face work occurred in most jobs, except for services considered essential, interruption of classes in the school environment, and closing of parks and gyms. Promoting an impact on daily life, not only through a probable decrease in the level of physical activity, but also in terms of the population’s diet [45].

The treatment of DM included diet, physical activity, insulin medication, and adequate self-control, so these changes promoted by the lockdown period could very likely have a negative impact on the treatment of individuals with DM; however, the studies in this review show better glycemic control [46].

Bonora et al., 2020 [38] observed that glycemic control improved in patients with T1DM monitored through the FSL, which stopped working during the lockdown, suggesting that the deceleration of routine daily activities may have beneficial effects on managing T1DM, at least in the short term. Eberle et al., 2021 [47] also found results similar to those of Bonora, demonstrating better glycemic control of individuals with type T1DM during the lockdown period through the use of various technological strategies for managing DM; these results may be associated with positive changes in self-care, such as a more strictly routine daily life, including scheduled meals and better medication management, as well as digital solutions aiding in glycemic control. However, D’Onofrio et al., 2021 [48] demonstrated deterioration in glucose homeostasis, specifically during the lockdown period, in terms of glycemic control in individuals with T2DM, monitored through face-to-face consultations. These differences can be explained by differences in management between the two types of DM; in T1DM, the treatment is more related to the control of drug therapy which, due to patients being at home, may have improved, while in T2DM, it Is mainly associated with lifestyle (where there was a decline in physical activity during this period).

Di Dalmazi et al., 2020 [37] reported that glycemic control metrics improved primarily in children and adults, while they remained unchanged in teens assisted by the FSL device during the COVID-19 lockdown period. Tinti et al., 2021 [49] showed an improvement in glucose metrics in children in the period of social isolation due to COVID-19, although the level of physical activity has decreased and insulin therapy must be adjusted, monitored through a glucose management sensor not specified in the study. In the case of children, it may be because their parents stayed home, and therefore had more time to manage their DM, better balancing their diet, exercise, and insulin needs to counteract the consequences of lockdown, such as physical inactivity and psychosocial impact, in addition to a lower level of stress related to the school environment. Tornese et al., 2020 [50], in contrast to Di Dalmazi et al., 2020 [37], described that T1DM glycemic control in adolescents using a hybrid system (Medtronic MiniMed™ 670 G) during restrictions due to the COVID-19 pandemic improved in those who continued physical activity during quarantine. In the case of Di Dalmazi et al., 2020 [37], not finding an improvement in glucose metrics in adolescents may have been due to the progressive distancing from the family that characterizes this period of life by the desire for independence and autonomy. However, in the study by Tornese et al., 2020 [50] even with this family distance, adolescents who continued physical activity demonstrated improved glycemic control. It is suggested that maintaining physical activity could compensate for the lack of monitoring carried out by parents during this period of life.

Navis et al., 2020 [39] described those glycemic results in adults with T1DM using FSL during the pandemic period improved. In agreement with Navis et al., 2020 [39], Aragona et al., 2020 [51] also reported a significant improvement in glycemic control with the Flash Glucose Monitoring FSL, in adult subjects using continuous monitoring during the lockdown period. These results are probably a reflection of the suspension of face-to-face work, promoting more regular activities of daily living and reducing work-related suffering.

Dover et al., 2020 [40] demonstrated that there was a small reduction in the number of subjects with hypoglycemia, and that blockade was not associated with a deterioration in glycemic control via the FSL device in people with T1DM using flash glucose monitoring. According to Samit Ghosal et al., (2020) [52], when daily glucose profiles were considered during the lockdown period, it was evident that most of the overall improvement in glycemic control was mainly due to a reduction in glucose levels in the blood in the early morning hours (4 a.m. to 10 a.m.). It was suggested that there was a less pronounced “dawn phenomenon”, that is, no need to use basal insulin. Although a clear explanation for this effect is not possible, the dawn phenomenon is associated with the release of counter-regulatory stress hormones. Thus, it can be hypothesized that a more regular lifestyle and lighter intelligent work activities may have reduced overall stress exposure and resulted in increased sleep quality and duration.

Cervantes-Torres and Romero-Blanco, 2022 [41] observed that the use of FSL during the lockdown was associated with a reduced number of hypoglycemic and hyperglycemic episodes in patients with T1DM, thus maintaining blood glucose levels within the ideal range to avoid acute and chronic complications. In addition, its use is related to better adherence to recommended habits in people with DM. The device also increased the frequency of self-monitoring of blood glucose by the sensor and avoided capillary punctures, making it easier to record data. It also improved dietary adherence and insulin administration. Navis et al., 2020 [39] also noted that glycemic outcomes in adults with T1DM using FSL during the pandemic period have improved.

Choudhary et al., 2022 [42] observed that through the FSL, it was possible to map the ups and downs of the DM in many individuals, better analyzing the time below the interval (TBR), the time above the interval (TAR) and time on Interval (TIR) in adults with the device during a block period, offering a more complete view of daily disease control. In agreement with this, Aragona et al., 2020 [51] reported a significant improvement in FSL glycemic control in adult individuals using continuous monitoring during the lockdown period.

In addition to the already-described factors related to a more regular lifestyle and a stricter daily routine, including scheduled meals, it was noted that digital treatments for T1DM patients (using control devices) likely had a positive impact on glycemic control. Some studies have already evaluated the effect of various digital therapy approaches for T1DM on health outcomes, and these treatments appear to be very promising. It is also important to point out factors that could be critical to the use of AI technology, such as (i) organizational structure, (ii) individuals’ lack of technical expertise, (iii) data governance issues, (iv) integration complexity, (v) low data quality, (vi) high cost of AI, (vii) confidentiality and security, and (viii) accountability and responsibility (Wirtz et al., 2019 [53]).

The current systematic review has some limitations. Although seven known databases were used, including more data sources could have improved the amount of literature included in the review. The same is true for search terms which, although inclusive, could have provided different results if a broader search strategy had been used, and therefore, not all relevant studies could have been identified. Furthermore, among the included studies, limitations are also present in terms of study design, heterogeneity of protocols for FSL, heterogeneity of control groups, cohorts, and the clear definition of the validation guideline to each of samples tested in the studies. This heterogeneity makes it very difficult to compare studies and interpret the effects of using FSL during blockade in diabetic subjects. Moreover, the results could be also stratified in consideration of the age of participants, as well as their DM type. 

The strength of this study is related to the relevance of the AI used in devices utilized for glycemic control in diabetic patients during the COVID-19 pandemic.

## 5. Facts and Perspectives

The aim was to demonstrate the relevance of AI in devices that facilitate blood glucose control during the COVID-19 pandemic lockdown period, on the basis of which it can be predicted that there will be an increase in the use of AI in devices that can aid in the monitoring of individuals with different diseases. Moreover, it is relevant to consider analytical results that do not have a constant value. It is verified that each result is characdterized by the properties of error and uncertainty. It is important to consider that the sources of these parameters must be known. The values of these parameters should be estimated. The analytical results, as they are obtained with AI, apply appropriate measurement methods. In addition, in future, it would desirable to perform a full comparison of the obtained results with data obtained using previously reported methods.

## 6. Conclusions

The use of FSL in patients with DM evidenced high patient compliance. This monitoring method has been shown to have a positive impact on glycemic control and on reducing the number of individuals with hypoglycemia. Therefore, the implementation of this sensor during COVID-19 confinement in this population seems possible, and appears to have been effective for observing the relevant parameters in diabetic patients. High-quality randomized controlled trials with adequate blinding are needed. Future studies should investigate the use of FSL in situations of social isolation, with adequate follow-up after the intervention, on relevant glycemic parameters, and should compare the effects of different types of AI and different brands, in order to determine the best protocol for these patients. The findings of the current systematic review suggest that the implementation of FSL during COVID-19 confinement in this population seemed to be confident and effective in people with diabetes mellitus. Nevertheless, the findings of the current systematic review must be interpreted with caution.

## Figures and Tables

**Figure 1 diagnostics-13-01499-f001:**
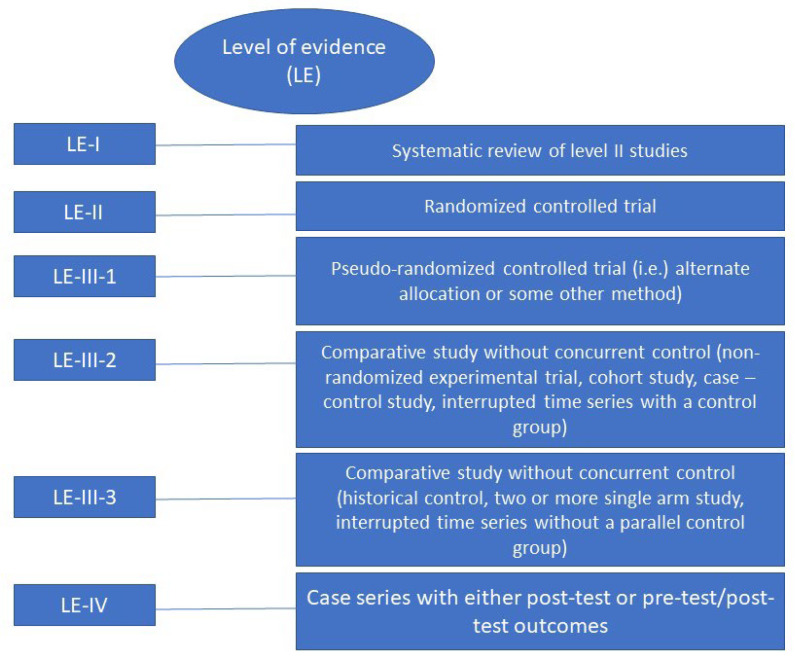
National Health and Medical Research Council evidence hierarchy scale.

**Figure 2 diagnostics-13-01499-f002:**
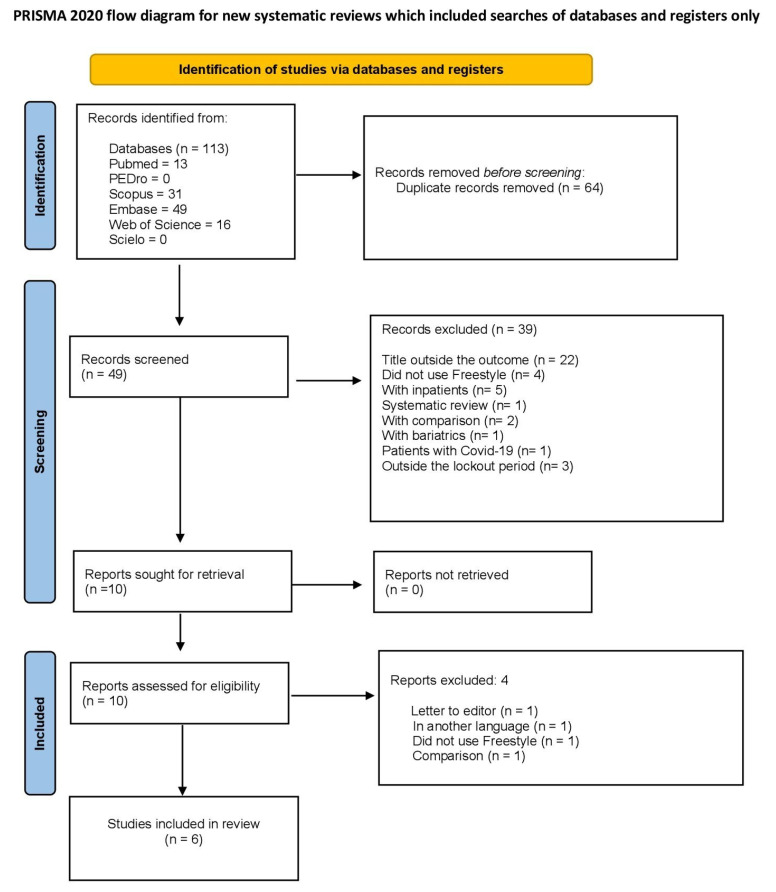
Flowchart with the steps of selection for this study.

**Figure 3 diagnostics-13-01499-f003:**
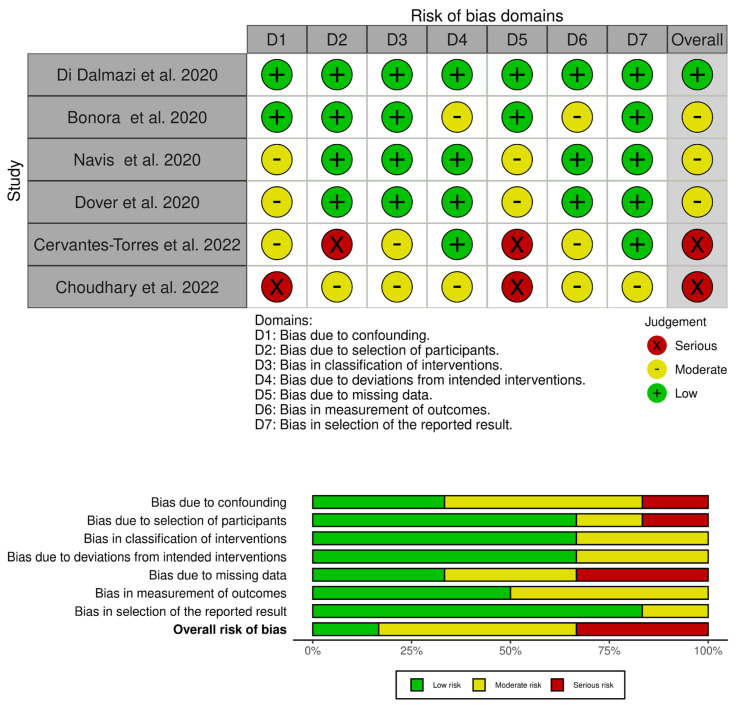
Risk of bias assessment of the studies included in this review [37,38,39,40,41,42].

**Table 1 diagnostics-13-01499-t001:** Main characteristics of the studies included.

Author/Year/Country	Anthropometric Characteristics of the Participants	Location	Period of Data Collection	Objective	Study Design	Level of Evidence
Di Dalmazi et al., 2020/Italy [37]	N = 130 with T1DM Children = 30 Teenagers = 24 Adults = 76 Male = 71 Female = 59	At home, telephonic contact	17 February 2020 and 5 April 2020, 20 days before and the 20 days after lockdown	To evaluate CGM metrics in children and adults with T1DM during lockdown and to identify the potentially related factors	Observational (Cohort)	III
Bonora et al., 2020/Italy [38]	N = 33 with T1DM Age = 36.9–45.0 years old Male = 12 Female = 21	DM outpatient clinic of the University Hospital of Padova; data were obtained using the FSL and sharing sensor data with the clinic on a web-based cloud system	At least 3 months	To examine glycemic control during the first week of lockdown against the spread of SARS-CoV-2 in people with T1DM during FGM in Italy in comparison to the pre-lockdown period	Observational (control case)	IV
Navis et al., 2020/UK [39]	N = 269 T1DM patients FSL = 190 DG6 = 79 Age = 41.4 ± 12.9 years old Male = 146 Female = 123	Large teaching hospital in the UK, data were obtained from outpatient electronic records and glucose monitoring web-based platforms on smarthphone device.	Pre-lockdown (1–14 February 2020), early lockdown (1–14 April 2020) and mid-lockdown (1–14 May 2020).	To assess whether sensor-based results before and during lockdown periods were different in a cohort of T1DM glucose sensor users.	Retrospective, observational, single-center cohort.	III
Dover et al., 2020/UK [40]	N = 572 T1DM Age = 38–39 years old Male = 301 Female = 271	Royal Infirmary of Edinburgh and Western General Hospital (Edinburgh) DM clinics; individuals linked their glucose data to these clinics using the LibreView platform.	March and May 2020.	To describe the effect of the lockdown on glycemic control in people with T1DM using flash glucose monitoring	Observational (Cohort).	III
Cervantes-Torres and Romero-Blanco, 2022/Spain [41]	N = 206 aged 46.6 ± 15.4 years old Male = 123 Female = 83	Data of type I diabetic patients were collected from a Spanish health area in the region of Castilla–La Mancha.	The first cut-off point was between February and March 2020 (before the use of FSL), and the second cut-off point was between February and March 2021 (after one year of using FSL).	To assess the effect of FSL device implantation in adult TiDM in the Health Area of Castilla–La Mancha (Spain) during the COVID-19 pandemic	Observational, cross-sectional, pre–post.	III
Choudhary et al., 2022/UK [42]	N = 8914 Aged 18 to ≥65 years old	Data were extracted from 8914 de-identified LibreView user accounts for adult users aged 18 years or older.	January to June 2020.	To evaluate the impact of the stay-at-home policy on different glucose metrics for time in range, time below range and time above range for FSL users within four defined age groups and on observed changes during the COVID-19 pandemic.	Observational	III

Legend: N = number of participants, T1DM = Type 1 diabetes mellitus; T2DM = Type 2 diabetes mellitus, FSL = FreeStyle Libre; DG6 = Dexcom G6; TAR = time above range, FGM = monitoring flash of glucose, UK = United Kingdom, HUS = Helsinki and Uusimaa, CSII = insulin pumps.

## Data Availability

The data presented in this study are available on request from the corresponding author.

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
