# Peer review of "Effectiveness of Using the FreeStyle Libre® System for Monitoring Blood Glucose during the COVID-19 Pandemic in Diabetic Individuals: Systematic Review"

_diagnostics, 2023, doi:10.3390/diagnostics13081499_

Round 1
Reviewer 1 Report
The four major components of the article are presented coherently and tightly linked. The list of bibliographic references is adequate; the documentation is appropriate regarding the titles consulted. This present work is well-designed, structured and developed in the laboratory. All methods applied to the samples meet the objective of the work. FreeStyle Libre® system benefits a better understanding of the whole process. This work must be disseminated among the scientific community, as it is an added value in Medical Science.
I suggest that the manuscript be acceptable after minor revision. Some detailed comments are as follows:
· Expand of Abstract based on analytical results and comparison.
· Also, including of the advantages and limitation of FreeStyle Libre® system on the analysis should be done in various sections of manuscript (abstract, introduction).
· what is the validation guideline for each of samples.
· Full comparison of obtained analytical results with previously reported methods.
· Expand of sections 5 and 6.
Author Response
Open Review
( ) I would not like to sign my review report
(x) I would like to sign my review report
Quality of English Language
( ) English very difficult to understand/incomprehensible
( ) Extensive editing of English language and style required
( ) Moderate English changes required
(x) English language and style are fine/minor spell check required
( ) I am not qualified to assess the quality of English in this paper
|
Yes |
Can be improved |
Must be improved |
Not applicable |
|
|
Does the introduction provide sufficient background and include all relevant references? |
( ) |
(x) |
( ) |
( ) |
|
Are all the cited references relevant to the research? |
(x) |
( ) |
( ) |
( ) |
|
Is the research design appropriate? |
(x) |
( ) |
( ) |
( ) |
|
Are the methods adequately described? |
( ) |
(x) |
( ) |
( ) |
|
Are the results clearly presented? |
(x) |
( ) |
( ) |
( ) |
|
Are the conclusions supported by the results? |
( ) |
(x) |
( ) |
( ) |
Comments and Suggestions for Authors
The four major components of the article are presented coherently and tightly linked. The list of bibliographic references is adequate; the documentation is appropriate regarding the titles consulted. This present work is well-designed, structured and developed in the laboratory. All methods applied to the samples meet the objective of the work. FreeStyle Libre® system benefits a better understanding of the whole process. This work must be disseminated among the scientific community, as it is an added value in Medical Science.
Thank you for your comments.
I suggest that the manuscript be acceptable after minor revision. Some detailed comments are as follows:
We thank you for your comments/criticisms. We tried to do our best to answer your comments. All the alterations and insertions are indicated in red in the new version of the manuscript.
Expand of Abstract based on analytical results and comparison.
It would be interesting to expand the abstract with your suggestion. But, unfortunately, there is a limitation of words in the abstract.
- Also, including of the advantages and limitation of FreeStyle Libre® system on the analysis should be done in various sections of manuscript (abstract, introduction).
t would be interesting to expand the abstract with your suggestion. However, unfortunately, there is a limitation of words in the abstract. But information about advantages and limitations of the AI and FreeStyle were added in the Introduction section.
- what is the validation guideline for each of samples.
We agree about the relevance, and we added comments in the limitations and in the perspectives of the current systematic review.
- Full comparison of obtained analytical results with previously reported methods.
We agree about the importance, and we added comments in the limitations and in the perspectives of the current systematic review.
- Expand of sections 5 and 6.
Surely, we agree, and your suggestions were followed.
Submission Date
15 March 2023

Reviewer 2 Report
Reviewer’s comments:
1. In ”Discussion” section, the systematic review for effectiveness of using FSL blood glucose monitoring discussed from many literatures including [37], [49], [45-48], and [50] should be compiled into a complete table to clearly reveal the treatment or therapy methods of T1DM and T2DM patients both 18-25 years age group and 65 years or older age group during COVID-19 pandemic.
2. The AI tool technique plays an important role for the effectiveness of glucose monitoring in FLS system, therefore the definition of the AI tool technique should be obviously exhibited by revealing the critical factor of AI system, why it can enhance the monitoring of diabetic patients during COVID-19 pandemic.
Author Response
Reviewer 2
Open Review
( ) I would not like to sign my review report
(x) I would like to sign my review report
Quality of English Language
( ) English very difficult to understand/incomprehensible
( ) Extensive editing of English language and style required
( ) Moderate English changes required
(x) English language and style are fine/minor spell check required
( ) I am not qualified to assess the quality of English in this paper
|
Yes |
Can be improved |
Must be improved |
Not applicable |
|
|
Does the introduction provide sufficient background and include all relevant references? |
( ) |
(x) |
( ) |
( ) |
|
Are all the cited references relevant to the research? |
(x) |
( ) |
( ) |
( ) |
|
Is the research design appropriate? |
(x) |
( ) |
( ) |
( ) |
|
Are the methods adequately described? |
(x) |
( ) |
( ) |
( ) |
|
Are the results clearly presented? |
( ) |
(x) |
( ) |
( ) |
|
Are the conclusions supported by the results? |
(x) |
( ) |
( ) |
( ) |
Comments and Suggestions for Authors
Reviewer’s comments:
We thank you for your comments/criticisms. We tried to do our best to answer your comments. All the alterations and insertions are indicated in red in the new version of the manuscript.
- In ”Discussion” section, the systematic review for effectiveness of using FSL blood glucose monitoring discussed from many literatures including [37], [49], [45-48], and [50] should be compiled into a complete table to clearly reveal the treatment or therapy methods of T1DM and T2DM patients both 18-25 years age group and 65 years or older age group during COVID-19 pandemic.
Thank you. Please, find in the limitations and in the perspectives of the study the information that you requested.
- The AI tool technique plays an important role for the effectiveness of glucose monitoring in FLS system, therefore the definition of the AI tool technique should be obviously exhibited by revealing the critical factor of AI system, why it can enhance the monitoring of diabetic patients during COVID-19 pandemic.
Thank you. Please, find in the limitations and in the perspectives of the study the information that you requested.
Submission Date
15 March 2023
